# The Effect of Group Identification on Death Anxiety: The Chain Mediation Role of Close Relationships and Self-Esteem

**DOI:** 10.3390/ijerph191610179

**Published:** 2022-08-17

**Authors:** Zilun Xiao, Yufang Zhao, Yingcan Zheng, Yan Bao, Chao Zhang

**Affiliations:** 1Department of Psychology, Southwest University, Chongqing 400715, China; 2Developmental Psychology for Armyman, Department of Medical Psychology, Army Medical University, Chongqing 400038, China

**Keywords:** group identification, death anxiety, self-esteem, close relationships

## Abstract

Based on the terror management theory (TMT), this study integrated self-esteem and close relationships to explore the effects of group identification on death anxiety. Five hundred and four participants completed the Death Anxiety, Rosenberg Self-Esteem, Social Identity, and Inclusion of Other in the Self scales via online platforms. There were significant correlations among group identification, close relationship, self-esteem, and death anxiety. Group identification had a significant negative predictive effect on death anxiety. Specifically, group identification affects death anxiety through two pathways: the separate mediating role of self-esteem and the serial mediation pathway of close relationships → self-esteem. Our study provides direct evidence that group identification relieves death anxiety. The results showed that the alleviating function of group identification was mediated by self-esteem and close relationships. This study provides a new perspective concerning TMT as a defense mechanism against death anxiety.

## 1. Introduction

Death anxiety refers to thoughts, fears, and emotions about death [1], which always emerge under life-threatening situations [2]. During the coronavirus (COVID-19) pandemic, more than 550 million people were confirmed to have contracted COVID-19. The death toll exceeded more than 6 million worldwide from Worldometer (https://www.worldometers.info/coronavirus/) accessed on June 2022. The rapid spread and high mortality rates induced fear and anxiety of death [3,4,5]. Death anxiety remained relatively high during the COVID-19 pandemic [6]. This anxiety decreases individual well-being and life satisfaction and increases fear, contributing to PTSD and other mental disorders [7,8]. Identifying ways to alleviate death anxiety and improve people’s mental health has become a public health issue requiring attention.

Reminders of death can encourage people to seek group identification and belonging. Empirical studies indicated that the salience of mortality leads to enhanced identification with meaningful ingroups such as country, gender, or university [9,10]. It encourages adherence to salient norms and values [11], prosocial behavior toward ingroup members [12], and support for ingroup leaders and their policies [13]. In the face of death, individuals often instinctively seek anxiety relief [1,14]. As mentioned above, when people face death, their sense of group identification increases. This phenomenon raises the question of whether stronger group identification can buffer death anxiety and, if so, how this process works. We will explore this question in depth in our article.

According to the terror management theory (TMT), throughout human history, people have constructed and identified a meaningful ingroup (for example, a country or society) to seek meaning in life and achieve symbolic immortality. Group identification arises from the need for cognitive consistency with the worldview of one’s group. Individuals with a strong ingroup identity feel that they belong to a group. Therefore, they have a stronger sense of symbolic immortality through the group [10,14,15]. By identifying with an ingroup (e.g., community, workplace, leisure activity club), people increase their connection to the world [16], enhance their symbolic immortality, and relieve their death anxiety [17,18]. Hence, from TMT’s perspective, people’s identification with the ingroup is closely related to its buffering against and alleviation of death anxiety.

TMT proposes that self-esteem serves as a defense mechanism against death anxiety [14]. Researchers believe that high self-esteem helps suppress death constructs after experiences of mortality salience [19,20,21]. It also reduces anxiety and concerns about death [18,22]. Numerous empirical studies have repeatedly confirmed self-esteem as a defense mechanism against death anxiety. The higher self-esteem is, the less death anxiety is experienced [20]. Group identification may alleviate death anxiety by promoting self-esteem. Previous researchers found that self-esteem originates from how people confirm their sense of worth and meaning. Ingroup identification is one way to accomplish this. [23]. Thus, group identity is an important source of self-esteem. The stronger group identification is, the higher self-esteem will be [24]. Self-esteem might also influence how group identification alleviates death anxiety because they are closely related. Therefore, we hypothesized that group identification alleviates death anxiety by providing a sense of value and meaning in life, with self-esteem mediating this relationship.

TMT proposes that close relationships with others are a defense mechanism against death anxiety [25]. We believe it may play an important role in group identification and anxiety about death. Earlier studies found that people sit closer to others after experiencing priming conditions related to mortality [26]. This response shows the importance of close relationships with others when facing death. Later studies have found that close relationships can relieve death anxiety. Raising children is an important way for adults to prolong life and fight symbolically against the inevitable [27,28,29]. These studies demonstrate that intimate relationships can provide peak experiences and help people feel fully alive [25], consequently reducing death anxiety. Furthermore, group identity includes not only cognitive coherence with group members but also includes establishing close emotional connections with them [25]. Thus, close relationships significantly influence group identification. Group identity leads to a more positive impression of the ingroup. It affects the individual’s awareness of ingroup cohesion [30]. Therefore, group identification may affect close relationships among its members. Previous studies also showed that close relationships with others, such as friends and parents [31,32], affect self-esteem. These studies show that close relationships provide more social support to individuals, leading to higher self-esteem. Therefore, it can be hypothesized that close relationships play a mediating role in the influence of group identification on self-esteem.

This study investigated the mitigating effects of group identification on death anxiety, based on TMT. It aimed to provide a perspective on how individuals can relieve death anxiety during the COVID-19 pandemic. We postulated the following hypothesis:

**Hypothesis** **1** **(H1):***Group identification negatively predicts death anxiety*.

There are two pathways by which group identification influences death anxiety:

**Hypothesis** **2a** **(H2a):**
*Self-esteem mediates the influence of group identification on death anxiety.*


**Hypothesis** **2b** **(H2b):**
*Close relationships and self-esteem play a chain mediating role between group identification and death anxiety.*


Figure 1 depicts the main hypothesized chain mediation model.

## 2. Methods

### 2.1. Participants

We recruited participants in China from an online panel platform: Credamo (https://www.credamo.com/, accessed on 28 October 2021). This online panel platform is designed for academic research and is considered reliable [33,34,35]. We received 505 responses, of which 504 remained after excluding an invalid response (i.e., asymmetric information before and after). In the final sample, the ages ranged from 17 to 69, with a mean age of 44.14 (SD = 12.28). The sample comprised and 52.98% female participants (*n* = 267) and 47.02% male participants (*n* = 237). Regarding the participants’ education, 12.5% had graduated from high school, 17.9% had junior college diplomas, 54% had Bachelor’s degrees, and 15.7% had Master’s degrees. Overall, 79.4% were married and 20.6% were unmarried. Students accounted for 6.94% of participants, employees of public institutions accounted for 45.16%, employees of enterprises accounted for 31.7%, self-employed accounted for 6.6%, and unemployed or retired accounted for 9.6%. The Human Research Ethics Committee of the authors’ institution approved the study. All participants were given the informed consent form. They were advised that they could freely withdraw from the study at any time. After the experiment was concluded and the data were deemed to be usable, each participant received 10 RMB in compensation.

### 2.2. Measures

#### 2.2.1. Death Anxiety

Death anxiety (e.g., “I am very afraid of death”) was measured with the 15-item Chinese version of the Templer Death Anxiety Scale (CT-DAS) [36]. Participants rated how much they disagreed or agreed with each item on a Likert scale ranging from 1 (totally disagree) to 5 (totally agree). The mean score was adopted, with higher scores indicating greater death anxiety. In this study, the scale showed high internal consistency (Cronbach’s α = 0.85).

#### 2.2.2. Self-Esteem

The 10-item Rosenberg Self-esteem Scale-Revised [37] was used to measure personal self-esteem. An example of a scale item is “Overall, I am satisfied with myself”. Participants rated the extent to which each statement most accurately described them on a 4-point rating scale (1 = not at all to 4 = very much). The mean score was adopted, with higher scores indicating greater self-esteem. The scale showed high internal consistency (Cronbach’s α = 0.86).

#### 2.2.3. Group Identification

The Social Identity Scale modified by Kwon and Lease (2009) [38] measured participants’ group identification, using items such as “I am glad that I am a member of my community or workplace”. Some participants were retired. Therefore, we divided them into two groups for evaluation: workplace and community (the latter group included those who had retired). Participants rated how much they disagreed or agreed with each item on a Likert scale ranging from 1 (totally disagree) to 7 (totally agree). The mean score was adopted, with higher scores indicating stronger identification with one’s group. The scale showed high internal consistency (Cronbach’s α = 0.94).

#### 2.2.4. Close Relationship with Group Members

The Inclusion of Other in the Self Scale (IOS) [39] was used to measure the degree of closeness (on a scale of 1–7) that participants felt toward other members of their community or workplace. The scale contains seven figures. In each figure, two circles represent the respondent and group members. The degree of overlap between the two circles represents how close the respondent feels to other group members.

### 2.3. Data Analysis

SPSS Version 22.0 software (IBM SPSS Statistics for Windows, Armonk, NY, USA: IBM Corp., 2014) was used to calculate descriptive statistics (e.g., frequencies, means, standard deviation) and to perform reliability analysis and correlation analysis. All models were adjusted for the demographic covariate of age.

Next, structural equation modeling using MPlus 8.3 examined the main hypothesized chain mediation model (group identification → close relationship → self-esteem → death anxiety). The following goodness of fit indices were used: Chi-square statistic $\chi^{2},$ $\chi^{2}$/df, RMSEA, CFI, TLI, SRMR.

## 3. Results

### 3.1. Common Method Bias

Exploratory factor analysis was used to test for common method bias [40]. First, we integrated all the study questionnaires. Next, exploratory factor analysis was carried out on all items. The interpretation of the probability of the first common factor was 25.57%, which is far less than 40% [41]. Therefore, there was no serious common method bias in the study data.

### 3.2. Correlation Analysis

Descriptive statistical analysis found that the 504 participants’ average death anxiety scores ranged from 1 to 4.73, their group identification scores ranged from 1 to 7, their close relationship scores ranged from 1 to 7, and their self-esteem ranged from 1.2 to 4.

Correlation analysis (Table 1) showed that group identification, close relationships, and self-esteem were significantly positively correlated with each other. Furthermore, there was a significant negative correlation for group identification, self-esteem, and close relationships, with death anxiety. Regression analysis showed that group identification had a significant negative predictive effect on death anxiety (b = −0.19, SE = 0.03, *p* < 0.001).

Significant correlations were found between age and group identification, close relationships, self-esteem, and death anxiety. Older individuals showed higher group identification, close relationships, and self-esteem, and lower death anxiety scores.

### 3.3. Testing the Main Hypothesized Chain Mediation Model

Significant correlations were found among group identification, close relationships, self-esteem, and death anxiety. Group identification had a significant negative predictive effect on death anxiety. This effect met the statistical requirements for a further indirect effect analysis. A test of self-esteem and close relationships as mediators between group identification and death anxiety yielded a good fit to the data $(\chi^{2}$/df = 1200/508 = 2.36, RMSEA = 0.05, CFI = 0.92, TLI = 0.91, SRMR = 0.08). The results of the test of the chain mediation effect showed that group identification positively predicts close relationships (r = 0.50, *p* < 0.001) and self-esteem (r = 0.43, *p* < 0.001). Close relationships positively predict self-esteem (r = 0.21, *p* < 0.001). Self-esteem negatively predicts death anxiety (r = −0.29, *p* < 0.001). Thus, the stronger group identification is, the more often closer relationships are experienced with individual group members. People feel that closer relationships with group members provide them with better self-esteem. Consequently, they report less death anxiety (see Figure 2).

Bootstrapping was performed to examine the mediation effects. Based on 1000 bootstrap samples with replacement, the 95% confidence interval (CI) of the indirect effect of “group identification → self-esteem → death anxiety” was [−0.15, −0.06]. The mediating effect accounted for 60.22% of the total effect. The confidence interval does not include zero, indicating that the mediating effect of self-esteem was statistically significant.

The 95% CI of the chain mediation effect of “group identification → close relationships → self-esteem → death anxiety” was [−0.05, −0.01]. The mediating effect accounted for 18.28% of the total effect. The confidence interval does not include zero, indicating that the chain mediating effect of close relationships and self-esteem was also statistically significant. However, the 95% CI of the indirect effect of “group identification → close relationships → death anxiety” was [−0.06, 0.03]. Thus, the mediating effect of close relationships was not significant.

## 4. Discussion

This study tested the alleviating function of group identification on death anxiety. Our results confirmed significant correlations among group identification, close relationships, self-esteem, and death anxiety. Group identification had a significant negative predictive effect on death anxiety. The stronger the group identification is, the more often close relationships are experienced with individual group members. These relationships provide individuals with better self-esteem, leading to lower death anxiety. This supports a chain mediating role between group identification and death anxiety.

Group identification’s role in relieving death anxiety may be related to a sense of belonging. Researchers found that enhancing the sense of belonging within a group can alleviate death anxiety [42]. Group identity is closely related to belonging, indicating the stronger the group identity, the greater the sense of group belonging [43]. Consequently, enhancing group identity can improve the sense of belonging and reduce self-uncertainty, which may alleviate death anxiety. In line with these associations, group identity could alleviate death anxiety. Participants who reported higher group identification claimed to have lower death anxiety.

The alleviating function of group identification on death anxiety depends on self-esteem’s mediation role. TMT proposes that humans realize the finiteness of life. However, the combination of survival instinct and mortality awareness creates the unique existence dilemma of humanity, resulting in death anxiety [10]. Positive self-esteem helps humans search for meaning in life, elevates symbolic immortality, and achieves ego transcendence. These factors provide protection against concerns about mortality [16,17]. Self-esteem is formed through the pursuit of cultural and group values [44]. The more strongly people identify with a group, the more they will shape their behaviors and concepts based on the group’s values, leading to higher self-esteem. Thus, group identification becomes an anxiety buffer through self-esteem.

Our results did not confirm that close relationships mediate between group identification and death anxiety. It is possible that the stimuli we selected for our study led to non-significant results, as they differed from previous studies using partners [45] or family [27,28,29]. We measured close relationships for group members, including colleagues or neighbors. Close relationships cause people to feel alive and resist death anxiety [25]. Thus, when individuals spend time with partners, family, or offspring, they experience stronger intimacy than with their colleagues or neighbors. According to our results, close relationships with colleagues are not sufficient to realize the fullness of life. They are less effective in defending against death anxiety. Furthermore, close relationships with colleagues or neighbors are social relationships. Building social relationships can enhance people’s self-esteem [46]. In our study, the social relationship network (close relationships) enhances people’s self-esteem and enables people to obtain a sense of meaning and value in life. Thus, forming close relationships with group members would improve people’s self-esteem and lower the experienced fear of death.

One interesting finding was that older people reported lower death anxiety scores than their younger counterparts. This result might be related to the sense of meaning in life. Compared with young people, older people have a stronger sense of the meaning of life [47]. Other researchers have found that a sense of life’s meaning can significantly negatively predict the elderly’s fear of death [48]. Individuals with a lower sense of life’s meaning experience more death anxiety, while individuals with a greater sense of life’s meaning are better able to accept mortality [49], which may explain why older people experience less death anxiety.

## 5. Implication and Future Directions

This study expands the idea of the TMT defense mechanism against death anxiety. Previously, researchers attributed the alleviating function of group identification on death anxiety to the worldview defense [9]. Our results provide a new path through self-esteem and close relationships as the alleviating function of group identification, which corresponds to TMT’s other two defense mechanism factors. Our study indicates that people with higher ingroup identification experience less death anxiety. This result means that increasing the connection and identification between individuals and their families and communities (such as schools, companies, or countries) reduces the threat of death and promotes greater well-being.

We cannot examine the causality of the association between age and death anxiety using this cross-sectional study. With a cross-sectional study, a wider range of data can be obtained in a shorter time. However, because of the short time span, the effect of time (age) might be neglected. Whether the changes in age over time predict changes in group identification and self-esteem and relieve death anxiety should be examined using a longitudinal design. Furthermore, we measured group identification only within one participant community or group. Individuals belong to multiple groups, and previous studies have shown that people with more social group memberships have better psychological well-being, are healthier, and live longer than those who belong to fewer [23,50]. The number and size of groups to which people belong are moderating variables for group identification in mitigating death anxiety. Thus, research investigating the role of multiple group memberships could provide comprehensive evidence regarding how people’s group identification reduces death anxiety. Moreover, other models may explain group identification’s alleviating mechanism on death anxiety. Group identification includes the close relationships established among members, which may regulate the effects of group identification on self-esteem. When individuals feel closer to other group members, the group has a greater value to them [30]. Thus, the sense of group identity is also important for the self-esteem formation process. Simultaneously, people have increased reliance on the group to feel protected against uncertainty and their fear of death. Therefore, future research could investigate the moderating effects of close relationships.

## 6. Conclusions

This study showed that there were significant correlations among group identification, close relationships, self-esteem, and death anxiety. Group identification had a significant negative predictive effect on death anxiety. Specifically, group identification affects death anxiety through two pathways: the separate mediating role of self-esteem and the serial mediation pathway of close relationships → self-esteem. Our results showed that the alleviating function of group identification was mediated by self-esteem and close relationships. This study provides a new perspective concerning TMT as a defense mechanism against death anxiety.

## Figures and Tables

**Figure 1 ijerph-19-10179-f001:**
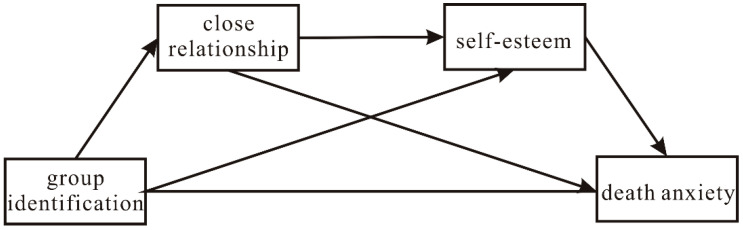
A conceptual model of group identification, close relationship, self-esteem, and death anxiety.

**Figure 2 ijerph-19-10179-f002:**
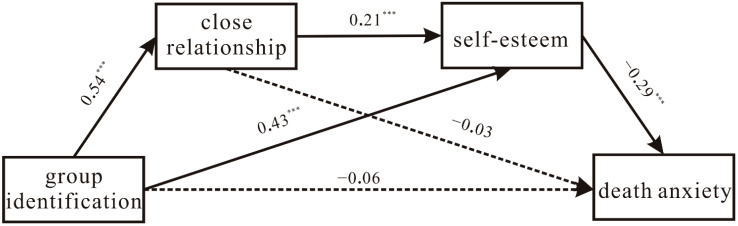
The chain mediation model of the impact of group identification on participants’ death anxiety. Note: Path values are standardized path coefficients. *** *p* < 0.001.

**Table 1 ijerph-19-10179-t001:** Bivariate correlations, means, and standard deviations of the variables (N = 504).

Variable	M	SD	(1)	(2)	(3)	(4)	(5)
(1) Group identification	5.52	1.10					
(2) Close relationship	4.86	1.24	0.57 **				
(3) Self-esteem	3.30	0.47	0.45 **	0.39 **			
(4) Death anxiety	2.89	0.65	−0.20 **	−0.20 **	−0.32 **		
(5) Gender			−0.03	−0.11 *	−0.01	0.13 **	
(6) Age	44.14	12.28	0.32 **	0.29 **	0.17 **	−0.16 **	−0.08

Note. Gender was coded as 1 = female, 2 = male. * *p* < 0.05; ** *p* < 0.01 (two-tailed Pearson correlation test).

## Data Availability

The data that support the findings of this study are available from the corresponding author upon reasonable request.

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
