# Peer review of "The Effect of Group Identification on Death Anxiety: The Chain Mediation Role of Close Relationships and Self-Esteem"

_ijerph, 2022, doi:10.3390/ijerph191610179_

Round 1

Reviewer 1 Report

The general research question (i.e., predictors of death anxiety) is a highly relevant one. Even though I find the topic interesting, I think the authors must make more efforts for this current manuscript to justify publication.

Title

The first part of the title suggests the idea of moderation whereas its second part brings the idea of serial mediation. As the title can be useful in communicating the study's purpose, I would suggest changing it.

Abstract

-          The mention of TMT should be placed at the beginning of the first sentence. In terms of study contributions, again, the buffering effect comes mainly from a moderator instead of a serial mediation.

Introduction

The introduction and the rationale for such a study are not convincing. My recommendation is that the introduction should be refined so that it is clear what you are investigating

-          What's the gap in the literature the study addresses? It needs to be clearer which conversation the authors wish to be part of, either in the literature on group identification or the literature on death anxiety, and what novel theoretical insights the research contributes in relation to this body of literature. I think that a clear theoretical perspective / conceptual framework should have been more explicitly applied. For example, Tajfel’s social identity theory is worth to be mentioned.

-          The theorizing underpinning the paths in the proposed mediating relationship is subject to very plausible reverse causation – please address it.

-          It would be helpful to have each hypothesis following the theoretical arguments and appropriately worded.

-          The definition of death anxiety originates before Ozeret al., 2021.  There are other sources to define death anxiety.  For example, Fortner BV, Neimeyer RA. Death anxiety in older adults: a quantitative review. Death Stud. 1999;23(5):387–411.

or Kesebir P. A quiet ego quiets death anxiety: humility as an existential anxiety buffer. J Pers Soc Psychol. 2014;106(4):610–23.

or Grant, A. M., & Wade-Benzoni, K. A. (2009). The hot and cool of death awareness at work: Mortality cues, aging, and self-protective and prosocial motivations. Academy of Management Review, 34, 600–622.

-          Figure 1 does not represent the serial mediation, but a parallel one.

Method

The age should be included as a control variable. I wonder if children (participants of 12) should have been included together with adults. In addition, the children need the parents’ approval and parents may be a source of bias in their evaluation. Also, self-esteem can be sensitive to other contextual indices in comparison to adults.

The age should be included as a control variable.

Discussion

The discussion is very much focused on the association between the study variables and some buffering effect with little explanation of the relationships.

The discussion in its current form seems to further confirm the impression that it is unclear what the key theoretical contribution is and how the stated contributions are derived from the empirical analysis, at least in the way it has been presented. This leaves the reader unconvinced about the analysis and the potential for a theoretical contribution.

 I would add that a detailed explanation of other limitations of the study would also be needed - for example, the cross-sectional approach of the data.

Reviewer 2 Report

I have completed my evaluation of a manuscript titled “Group identification alleviates death anxiety: The chain mediation role of close relationships and self-esteem

This work raises some serious concerns that must be addressed.

Title

If the aim of this study is to provide a perspective on how individuals can relieve death anxiety during the COVID-19 pandemic the authors should consider this term in the title. In my opinion, the author needs to add “COVID-19” in the title

Abstract

If the aim of this study is to provide a perspective on how individuals can relieve death anxiety during the COVID-19 pandemic the authors should consider this term in the abstract!

Introduction

If the aim of this study is to provide a perspective on how individuals can relieve death anxiety during the COVID-19 pandemic the author should better explain the reason for this aim (I suggest after 1st paragraph of the introduction)

Participant

The participants described very insufficiently. In Line 105-106 authors write “The sample comprised 46.9% male participants (n=237)” This is very confusing because the final sample seems to consist of only male respondents. Therefore, it would be desirable also to add how many women there were in the final sample.

There is no information about participants, for example, who they are, did they have comorbidities, what is their working profession, or if their health professionals among them…How do the authors recruit participants? From where? Which were the inclusion and exclusion criteria? I'm missing a better description of participants, or a table with descriptions of respondents

Methods

The authors described the Rosenberg Self-esteem Scale-Revised but in the abstract written Beck Rosenberg Self-esteem Scale-Revised?

Data Analysis

Very deficient, because it only contains data about the names of statistical programs, used by the authors. There is no description of statistical methods, description, data distribution, etc. Why did the authors have the main model and an alternative model? It is not entirely clear what is gained by this, so it is necessary to explain it better in the data analysis. The models are not described. Why did the authors use Mplus program, what did they want to identify, and in what way?

The authors mention the chain mediation model. This model also needs to be explained in statistical methods. All this should be described in the data analysis.

Additionally, in chapter 3.4. , the authors write "Other models may exist to explain the buffering mechanism of group identification on death anxiety. " This sentence needs to be in chapter data analysis, not in results.

 Results

Common Method Bias

Exploratory factor analysis was used to test common method bias (Zhou, Long, 2004). Why is this reference missed in the reference list)

Correlation analysis

Table 1 needs an explanation in the footnotes. The table is very insufficient. Which the test of correlation test was used (Pearson or Spearman) and whether data in table r-coefficient or p-value. Also, the values in the table must be complete (for example zero before the comma is missing. (for example   .33 - it is necessary to write 0.33)

Test of alternative models

Figure 2. Is the number on the arrow line are the strength of correlation or p-value. Why on the bottom of the picture the p-value with two decimal places and then with three? P-value is always written with three decimal places

Figure 3. and Figure 4. The author should give clarification at the bottom of the picture and write the number (results) on the arrow line. The other comments as well as Figure 2. There are no results on the arrow line. Clarification is requested at the bottom of the picture

If these pictures are an illustration of the alternative model then it should be described and replaced from this place, and put in the methods (to describe the alternative model). If it is no illustration, then needs to contain the results on the arrow line. This is very confusing.

Again, why these alternative models. Please explain in methods!

Discussion

The author did not explain associations between their results and COVID-19. They give explanations about results but without an overview on Covid -19

Round 2

Reviewer 1 Report

The revision has strengthened the paper mostly in its theoretical approach. So, the manuscript addresses the mediation role of close relationship and self-esteem in the relationship between group identification and death anxiety. However, there are still things to be addressed in the light of the revised theoretical framework. 

The main issue I see now consists of the alternative models. Given the fact that the authors have already chosen the mediating role of the close relationship and supported the theoretical design in that vein, I wonder why it would be necessary to test the alternative models. The alternative models should be proposed as future research directions. 

The explanation on education level as control variable is missing. 

At page 5 - The bivariate correlations are reported when explain the mediation model. Also, there are some errors in reporting r coefficients - for example, the correlation between close relationship and self-esteem is .54 in text, but .57 in table 1. I suggest presenting the bivariate correlations in the section 3.2. Of course, careful verification of the indices is needed. 

Furthermore, a table with the results of regression analysis would help to better understand the data. Please verify the percentage of 60.22%. Isn't it explained variance? 

Figure 1 contains the bivariate indices. There should be the regression coefficients.  

Section 3.4 - I don't see it necessary. 

In the discussion section, I suggest avoiding the word "buffer" - the type of the research and the way the data was tested can not imply such an effect. 

page 7 - lines 269-70. "Participant who exclaimed" - please revised 

page 7 - lines 282-296. The discussion of the mediating role of close relationship has not been hypothesised. Consequently, there is no need to discuss it.

page 7 - lines 297 - 304 - these discussions should be prompted by results reported in the previous section. 

Reviewer 2 Report

I have completed my reevaluation of a manuscript titled “The effect of group identification on death anxiety: The chain mediation role of close relationships and self-esteem

This is a much-improved version, so the manuscript can be accepted in its present form.

Author Response

Thank you so much for your kind words and supportive comments! Your feedback has improved our manuscript. We greatly appreciate your careful review of our manuscript.
